# Inhibition of ANO1 by *Cis*- and *Trans*-Resveratrol and Their Anticancer Activity in Human Prostate Cancer PC-3 Cells

**DOI:** 10.3390/ijms24021186

**Published:** 2023-01-07

**Authors:** Dongkyu Jeon, Minjae Jo, Yechan Lee, So-Hyeon Park, Hong Thi Lam Phan, Joo Hyun Nam, Wan Namkung

**Affiliations:** 1College of Pharmacy and Yonsei, Institute of Pharmaceutical Sciences, Yonsei University, 85 Songdogwahak-ro, Yeonsu-gu, Incheon 21983, Republic of Korea; 2Department of Physiology, College of Medicine, Dongguk University, 123 Dongdae-ro, Gyeongju 38066, Republic of Korea; 3Channelopathy Research Center (CRC), College of Medicine, Dongguk University, 32 Dongguk-ro, Ilsan Dong-gu, Goyang 10326, Republic of Korea

**Keywords:** *cis*-resveratrol, *trans*-resveratrol, anoctamin 1, inhibitor, anticancer

## Abstract

Anoctamin1 (ANO1), a calcium-activated chloride channel, is involved in the proliferation, migration, and invasion of various cancer cells including head and neck squamous cell carcinoma, lung cancer, and prostate cancer. Inhibition of ANO1 activity or downregulation of ANO1 expression in these cancer cells is known to exhibit anticancer effects. Resveratrol, a natural polyphenol abundant in wines, grapes, berries, soybeans, and peanuts, shows a wide variety of biological effects including anti-inflammatory, antioxidant, and anticancer activities. In this study, we investigated the effects of two stereoisomers of resveratrol on ANO1 activity and found that *cis*- and *trans*-resveratrol inhibited ANO1 activity with different potencies. *Cis*- and *trans*-resveratrol inhibited ANO1 channel activity with IC_50_ values of 10.6 and 102 μM, respectively, and had no significant effect on intracellular calcium signaling at 10 and 100 μM, respectively. In addition, *cis*-resveratrol downregulated mRNA and protein expression levels of ANO1 more potently than *trans*-resveratrol in PC-3 prostate cancer cells. *Cis*- and *trans*-resveratrol significantly reduced cell proliferation and cell migration in an ANO1-dependent manner, and both resveratrol isomers strongly increased caspase-3 activity, PARP cleavage, and apoptotic sub-G1 phase ratio in PC-3 cells. These results revealed that *cis*-resveratrol is a potent inhibitor of ANO1 and exhibits ANO1-dependent anticancer activity against human metastatic prostate cancer PC-3 cells.

## 1. Introduction

Anoctamin1 (ANO1), also known as transmembrane protein 16A (TMEM16A), has been identified as a calcium-activated chloride channel [1,2,3]. ANO1 is expressed broadly in various tissues and plays an important role in diverse physiological functions including transepithelial fluid secretion, smooth muscle contraction, neuronal excitation, and cell proliferation [4,5,6,7,8]. In particular, the *ANO1* gene is located within the 11q13 amplicon which is frequently amplified in many malignant tumors and ANO1 has been reported to be highly expressed in a variety of human cancers including gastrointestinal stromal tumor, head and neck squamous cell carcinoma, prostate cancer, breast cancer, pancreatic cancer, and glioblastoma [9,10,11,12,13,14,15]. In addition, the increased expression of ANO1 is known to be associated with poor prognosis in cancer patients [16].

Even though the underlying mechanism is not clear, inhibition of ANO1 activity and downregulation of ANO1 expression show anticancer effects by inhibiting cell proliferation, migration, and invasion and inducing apoptosis in various cancers [12,17,18,19,20]. For instance, ANO1 knockdown with small hairpin RNAs (shRNAs) inhibited the proliferation, migration, and invasion of human lung cancer cells and the tumor growth was significantly reduced by ANO1 silencing in nude mice [21]. A significant reduction in proliferation, metastasis, and invasion of human metastatic prostate cancer PC-3 cells was observed after treatment with shRNA targeting human ANO1. Furthermore, tumor growth was significantly reduced by intratumoral injection of ANO1 shRNA in an orthotopic xenograft mice model using PC-3 cells [22]. To date, several compounds showing ANO1 inhibition and/or ANO1 downregulation have been reported to show anticancer effects, including T16A_inh_-01 [17], idebenone [18], luteolin [23], Ani9-5f [24], Ani-D2 [25], cinobufagin [19], and diethylstilbestrol [26]. In particular, the anticancer effects of ANO1 inhibition have been well-established in human prostate cancer cells. Pharmacological blockage of ANO1 activity with idebenone reduced cell proliferation and induced apoptosis in PC-3 cells [18]. Downregulation of ANO1 expression by luteolin and Ani-D2 also showed a reduction in cell proliferation and migration and induction of apoptosis in PC-3 cells [23,25]. Upregulation of TNF-α signaling was reported to be involved in inducing apoptosis by ANO1 downregulation in PC-3 pancreatic cancer cells [27].

On the other hand, activation of the ANO1 channel was also suggested to have therapeutic potential in human diseases including salivary gland dysfunction, cystic fibrosis, dry eye syndrome, and intestinal hypomotility [28]. Since the identification of ANO1, only a few selective ANO1 activators and potentiators such as E_act_ and F_act_ have been reported [29]. Interestingly, it was recently reported that resveratrol can also activate ANO1 chloride channel [30].

Resveratrol, a trihydroxylated stilbene (3,5,4′-trihydroxystilbene), is a naturally occurring polyphenol found in significant amounts in wines, grapes, berries, soybeans, and peanuts as well as in Chinese and Japanese herbal medicines [31,32]. Resveratrol initially attracted little interest until it was postulated to be responsible for the so-called ‘French Paradox’ in 1992 [33]. Since then, resveratrol has been reported to show a wide variety of biological effects including antioxidant, anti-aging, anti-inflammatory, anticancer, cardioprotective, and anti-diabetic activities [34]. Especially, anticancer activities of resveratrol have attracted many researchers since its cancer chemopreventive activity was first reported in 1997 [35]. Resveratrol shows anticancer effects in a wide variety of tumor cells including breast, prostate, pancreatic, stomach, and colorectal cancers, head and neck squamous cell carcinoma, ovarian carcinoma, and hematological malignancies [32,36,37]. Resveratrol is known to exert its anticancer activities in various cancer states including initiation, promotion, and progression by affecting the diverse signal pathways [38].

Resveratrol is present in two isomeric forms, *cis*- and *trans*-resveratrol, with *trans*-isomer being more predominant and stable [39]. Exposure of *trans*-resveratrol to solar or ultraviolet radiation can induce *cis*-isomerization [40,41]. Although *cis*-resveratrol has never been detected in fresh grapes, a mixture of *cis*- and *trans*-resveratrol isomers exists in wines [42]. Until recently, *trans*-resveratrol was the main focus of investigation and most of the observed health benefits can be attributed to *trans*-resveratrol. Even though *cis*-resveratrol has not been well-explored, several studies have reported its biological activities including anti-inflammatory, antioxidant, and anticancer effects [43,44].

In this study, we investigated the effects of the two isomers of resveratrol, *cis*- and *trans*-resveratrol, on ANO1 channel activity and unexpectedly found that two resveratrol isomers inhibited ANO1. Since ANO1 inhibitors show anticancer effects, *cis*- and *trans*-resveratrol were further investigated on their anticancer activities in human prostate cancer PC-3 cells.

## 2. Results

### 2.1. Inhibitory Effect of Cis- and Trans-Resveratrol on ANO1 Channel Activity

To investigate the effect of two stereoisomers of resveratrol, *cis*- and *trans*-resveratrol, on ANO1 chloride channel activity, measurement of apical membrane currents was performed in Fisher rat thyroid (FRT) cells expressing human ANO1. As shown in Figure 1A–D, the application of *cis*-resveratrol and *trans*-resveratrol did not induce ANO1 activation, but rather inhibited the activation of ANO1 by ATP in a dose-dependent manner. Interestingly, *cis*-resveratrol blocked ANO1 channel activity more potently compared to *trans*-resveratrol. A total of 10 and 30 μM of *cis*-resveratrol inhibited ANO1 activity by 52 and 86%, respectively, and 100 and 300 μM of *trans*-resveratrol inhibited ANO1 activity by 31 and 66%, respectively. To evaluate IC_50_ values for *cis*- and *trans*-resveratrol, we performed YFP fluorescence quenching assays in YFP-F46L/H148Q/I152L, a halide sensor, and human ANO1-expressing FRT cells. As shown in Figure 1E–G, both *cis*- and *trans*-resveratrol significantly inhibited ANO1 activity in a dose-dependent manner, and the ATP-induced decrease in YFP fluorescence was completely blocked by Ani9, a potent and selective ANO1 inhibitor [45]. The IC_50_ value of *cis*- and *trans*-resveratrol was 10.6 μM and 102 μM, respectively.

To confirm the inhibitory effect of *cis*- and *trans*-resveratrol on ANO1 chloride channel activity, whole-cell patch clamp experiments were performed in ANO1-overexpressing HEK293T cells. As shown in Figure 2, *cis*- and *trans*-resveratrol significantly inhibited ANO1 currents. *Cis*-resveratrol inhibited ANO1 chloride currents by 23% and 60% at 10 and 30 μM, respectively. *Trans*-resveratrol inhibited ANO1 chloride currents by 47% and 80% at 100 and 300 μM, respectively, more weakly than *cis*-resveratrol (Figure 2C,F).

### 2.2. Effect of Cis- and Trans-Resveratrol on Intracellular Calcium Levels

Intracellular calcium signaling plays an essential role in the regulation of ANO1 channel activity. To further investigate the mechanism of ANO1 inhibition by *cis*- and *trans*-resveratrol, the effect of both resveratrol isomers on intracellular calcium levels was measured using a fluorescent calcium indicator Fluo-4. As shown in Figure 3A,B, the application of high concentrations of *cis*- and *trans*-resveratrol did not alter the intracellular calcium levels, but ionomycin, a calcium ionophore, strongly increased intracellular calcium levels. In addition, pretreatment of 10 μM *cis*-resveratrol and 100 μM *trans*-resveratrol did not significantly affect the ionomycin-induced increase in intracellular calcium levels. However, high concentrations of *cis*-resveratrol (30 μM) and *trans*-resveratrol (300 μM) showed partial inhibition (Figure 3C,D).

### 2.3. Effects of Cis- and Trans-Resveratrol on Protein and mRNA Expression Levels of ANO1

Several ANO1 inhibitors including luteolin and Ani9-5f reduce protein expression levels of ANO1 [23,45]. To investigate the effect of *cis*- and *trans*-resveratrol on protein expression levels of ANO1, Western blot analysis was performed in PC-3 cells expressing high levels of ANO1. *Cis*-resveratrol significantly reduced ANO1 protein expression in a dose-dependent manner, and *trans*-resveratrol also significantly reduced protein expression levels of ANO1 (Figure 4A–D). Notably, 30 μM *cis*-resveratrol and 300 μM *trans*-resveratrol had comparable effects on the reduction of ANO1 protein levels by 10 μM Ani9-5f. To further investigate whether *cis*- and *trans*-resveratrol affect the mRNA expression level of ANO1, real-time PCR analysis was performed in PC-3 cells. As shown in Figure 4E, 10 μM *cis*-resveratrol and 100 μM *trans*-resveratrol had no effect on ANO1 mRNA expression, whereas high concentrations of *cis*-resveratrol (30 μM) and *trans*-resveratrol (300 μM) significantly reduced mRNA expression levels of ANO1.

### 2.4. Inhibitory Effects of Cis- and Trans-Resveratrol on Cell Proliferation and Migration in PC-3 Cells

Resveratrol has been extensively researched on its anticancer activity in various tumor cells [32,36,37]. Furthermore, pharmacological inhibition and downregulation of ANO1 showed anticancer effects in metastatic prostate cancer cells [23,27]. To investigate the effects of *cis*- and *trans*-resveratrol on cell proliferation, cell viability was measured in the highly metastatic prostate cancer PC-3 cells. *Cis*-resveratrol significantly reduced cell viability of PC-3 cells by 17, 45, 97, and 98% at 3, 10, 30, and 100 μM, respectively (Figure 5A). *Trans*-resveratrol had a less potent inhibitory effect on cell proliferation. Cell viability was significantly reduced by 22, 30, and 56% at 10, 30, and 100 μM of *trans*-resveratrol, respectively (Figure 5B).

A wound healing assay was performed to observe the effects of *cis*- and *trans*-resveratrol on cell migration in PC-3 cells. As shown in Figure 5C,D, *cis*- and *trans*-resveratrol significantly reduced cell migration in a dose-dependent manner.

### 2.5. Cis- and Trans-Resveratrol Induce Apoptosis in PC-3 Cells

To investigate whether *cis*- and *trans*-resveratrol induce apoptosis, we observed caspase-3 activity and PARP cleavage in PC-3 cells. As shown in Figure 6A,B, *cis*- and *trans*-resveratrol significantly increased caspase-3 activity and 10 μM *cis*-resveratrol showed stronger caspase-3 activation compared to 100 μM *trans*-resveratrol. The *cis*-resveratrol-induced increase in caspase-3-positive cells and caspase-3 activity was completely blocked by Ac-DEVD-CHO, a synthetic tetrapeptide inhibitor of caspase-3/7. In the case of PARP cleavage, *cis*-resveratrol and *trans*-resveratrol significantly increased PARP cleavage in PC-3 cells in a dose-dependent manner (Figure 6C).

To observe the effect of *cis*- and *trans*-resveratrol on the cell cycle of PC-3 cells, flow cytometry analysis using propidium iodide was performed. As shown in Figure 7, *cis*- and *trans*-resveratrol strongly increased the ratios in the sub-G1 (apoptotic peak) phase in a dose-dependent manner. Notably, 30 μM *cis*-resveratrol induced a decrease in the G0/G1 phase from 84.6% to 62.7% and an increase in the sub-G1 phase from 8.2% to 28.5%. In the case of 300 μM *trans*-resveratrol, the G0/G1 phase was strongly decreased from 78.1% to 42.9% and the sub-G1 phase was increased from 13.0% to 44.1%. Interestingly, 30 μM *cis*-resveratrol did not affect the G2/M phase ratio, whereas 300 μM *trans*-resveratrol increased the G2/M phase ratio from 3.3% to 8.0%. These results suggest that, unlike *trans*-resveratrol, *cis*-resveratrol induces apoptosis without significantly affecting cell cycle arrest.

## 3. Discussion

Several ANO1 modulators have been reported since it was identified as a calcium-activated chloride channel in 2008 [46]. ANO1 inhibition has shown therapeutic potential as an anticancer, anti-asthmatic, and antinociceptive agent [16,46]. Potentiation of ANO1 can provide health benefits for human diseases such as salivary gland dysfunction, cystic fibrosis, dry eye syndrome, and intestinal hypomotility [29]. Even though several selective ANO1 inhibitors have been identified, only a few selective ANO1 activators have been reported [46]. Interestingly, resveratrol, a natural polyphenol found in wines, grapes, and berries, has recently been reported as an activator of ANO1 [30]. Unexpectedly, however, we found that *cis*-resveratrol, in particular, strongly inhibited ANO1. As shown in Figure 1A–D, both *cis*- and *trans*-resveratrol treatment did not induce an increase in ANO1 chloride currents in FRT cells expressing human ANO1. On the contrary, pretreatment of *cis*- and *trans*-resveratrol inhibited the ATP-induced increase of ANO1 chloride currents in a dose-dependent manner. The whole-cell patch clamp study also showed that both *cis*- and *trans*-resveratrol potently inhibited ANO1 chloride currents in ANO1-expressing HEK293T cells in a dose-dependent manner (Figure 2). These electrophysiological studies clearly showed that *cis*- and *trans*-resveratrol are ANO1 inhibitors rather than ANO1 activators. It is difficult to clearly explain the reason why contradictory results were observed. Interestingly, however, a previous study reported that resveratrol increased Cl^−^ secretion in porcine ileum but significantly inhibited Ca^2+^-induced Cl^−^ secretion [47]. The previous study suggested that the activation of cystic fibrosis transmembrane conductance regulator (CFTR) may be responsible for resveratrol-induced Cl^−^ secretion, and resveratrol may inhibit Ca^2+^-induced Cl^−^ secretion. Taken together, these results suggest that ANO1 inhibition by resveratrol may be involved in the inhibition of Ca^2+^-induced Cl^−^ secretion by resveratrol observed in porcine ileum.

In order to investigate the mechanism of action of *cis*- and *trans*-resveratrol to inhibit ANO1, intracellular calcium signals were observed. *Cis*- and *trans*-resveratrol had minimal effects on calcium signals at concentrations up to 10 μM and 100 μM, respectively (Figure 3). On the other hand, *cis*- and *trans*-resveratrol strongly inhibited ANO1 at concentrations of 10 μM and 100 μM, respectively (Figure 1). Although we could not elucidate the reason for the different ANO1 inhibitory potency of the two stereoisomers of resveratrol, we found that they did not significantly interfere with calcium signaling and mRNA expression of ANO1 at a concentration caused a significant decrease in ANO1 channel activity and ANO1 protein levels (Figure 3 and Figure 4). These results suggest that the ANO1 inhibitory effects of *cis*- and *trans*-resveratrol may be caused by the direct binding of the two isomers of resveratrol to ANO1 with different binding modes.

ANO1 has been suggested as a therapeutic target for various cancers since it is highly amplified or overexpressed in various types of cancer and pharmacological inhibition or downregulation of ANO1 shows anticancer activities [12,13,14]. In previous studies, ANO1 inhibitors with significant inhibitory effects on channel activity as well as protein expression showed strong inhibition of cell viability and migration in PC-3 cells expressing high levels of ANO1 [18,23,24,25,26]. It was suggested that the reduction in ANO1 protein levels might be more critical in demonstrating anticancer effects than the inhibition of ANO1 channel activity [25]. In the present study, we found that *cis*- and *trans*-resveratrol directly inhibited ANO1 channel activity with a decrease in ANO1 protein levels (Figure 4). Interestingly, *cis*- and *trans*-resveratrol also showed inhibitory effects on cell proliferation and migration in PC-3 prostate cancer cells (Figure 5). In addition, *cis*- and *trans*-resveratrol significantly promoted caspase-3 activation and PARP cleavage and increased the apoptotic sub-G1 phase ratio in PC-3 cells (Figure 6 and Figure 7).

Resveratrol suppressed the growth of prostate cancer via the down-regulation of androgen receptor (AR) expression in the transgenic adenocarcinoma mouse prostate model [48]. In addition, resveratrol reduced the volume of tumors by lowering tumor-cell proliferation and neovascularization and inducing apoptosis in xenograft mice models of the AR-negative PC-3 cell [49]. Even though the majority of research on resveratrol’s anticancer effects has been focused on *trans*-isomer, *cis*-resveratrol has also been reported to show anticancer effects [50]. *Trans*-resveratrol has been reported to show stronger anticancer effects than the *cis*-isomer. For example, in human colorectal tumor SW480 cells, *trans*-resveratrol showed antiproliferative activities with the IC_50_ of 20 ± 3 μM, whereas the IC_50_ value of *cis*-resveratrol was 90 ± 12 μM [51]. *Trans*-resveratrol also exhibited slightly stronger cytotoxic activities than the *cis*-isomer in pancreatic cancer, breast cancer, lung small cell cancer, colon cancer, and prostate cancer cell lines [52]. However, in the present study, *cis*-resveratrol showed stronger anticancer effects than *trans*-resveratrol in PC-3 cells expressing high levels of ANO1, which is in correlation with their effects on the inhibition of ANO1 channel activity and the reduction in ANO1 expression. These results indicate that the anticancer mechanism of resveratrol, especially *cis*-resveratrol, may involve, at least in part, the inhibition and downregulation of ANO1.

In many clinical trials, resveratrol is well-tolerated and pharmacologically safe up to 5 g/day [53]. However, *trans*-resveratrol is rapidly metabolized mainly by sulfate and glucuronic acid conjugation in the intestine and liver and rapidly excreted [54]. After a single 500 mg oral dose in healthy volunteers, C_max_ was measured as 71.2 ng/mL, 4083.9 ng/mL, and 1516.0 ng/mL for resveratrol, glucuronated resveratrol, and sulphated resveratrol, respectively, and no side effects related to resveratrol were reported during the study [55]. However, comprehensive pharmacokinetic data on *cis*-resveratrol are not yet available. Therefore, in order to develop *cis*-resveratrol as an anticancer agent, various studies including pharmacokinetic studies should be conducted in the future.

In conclusion, *cis*- and *trans*-resveratrol inhibited ANO1 channel activity and down-regulated mRNA and protein expression levels of ANO1. Notably, *cis*-resveratrol exhibited a stronger inhibition of ANO1 activity than *trans*-resveratrol and a stronger cell growth and migration inhibitory effect in PC-3 prostate cancer cells. These results suggest a new therapeutic potential for *cis*-resveratrol in cancers with high ANO1 expression.

## 4. Materials and Methods

### 4.1. Cell Culture

Fisher rat thyroid (FRT) cells stably expressing ANO1 were provided by Alan Verkman (University of California, San Francisco, CA, USA). ANO1-expressing FRT cells were stably transfected with the halide sensor YFP-H148Q/I152L/F46L as described in a previous study [56]. FRT cells were cultured in Ham’s F-12 Modified medium with 10% fetal bovine serum (FBS), 100 units/mL penicillin, 100 μg/mL streptomycin, and 2 mM L-glutamine. PC-3 cells were purchased from Korean Cell Line Bank (KCLB) and cultured in RPMI 1640 medium supplemented with 10% FBS, 100 units/mL penicillin, and 100 μg/mL streptomycin. FRT cells and PC-3 cells were grown at 37 °C, 5% CO_2_, and 95% humidity. HEK293T cells were purchased from the American Type Culture Collection and cultured in Dulbecco’s modified Eagle’s medium supplemented with 10% FBS and 1% penicillin/streptomycin at 37 °C with 10% CO_2_.

### 4.2. Materials

*Cis*- and *trans*-resveratrol were purchased from Tocris Bioscience (Bristol, UK). All other materials not described were purchased from Sigma-Aldrich (St. Louis, MO, USA).

### 4.3. Ussing Chamber Assay

ANO1-expressing FRT cells were cultured on snapwell inserts (1.12 cm^2^ surface area) until confluent. Then snapwell inserts were mounted in Ussing chambers (Physiologic Instruments, San Diego, CA, USA). The basolateral side chamber was bathed with HCO_3-_-buffered solution containing (in mM) 120 NaCl, 5 KCl, 1 MgCl_2_, 1 CaCl_2_, 10 D-glucose, 2.5 HEPES, and 25 NaHCO_3_ (pH 7.4). The apical side chamber was filled with a half-Cl^−^ solution in which 65 mM NaCl was replaced by Na-gluconate. Cells were bathed for 10 min while being aerated with 95% O_2_ and 5% CO_2_ at 37 °C. *Cis*- and *trans*-resveratrol were applied to both apical and basolateral bath solutions 10 min before ATP application to the apical bath solution to activate ANO1. Membrane currents were measured and recorded with a 4 Hz sampling rate using an EVC4000 Multi-Channel V/I Clamp (World Precision Instruments, Sarasota, FL, USA) and PowerLab 4/35 (AD Instruments, Castle Hill, Australia). Data were collected and analyzed with Lab Chart Pro 7 (AD Instruments, Castle Hill, Australia).

### 4.4. YFP Fluorescence Quenching Assay

FRT cells that stably express both ANO1 and YFP variant were incubated in 96-well microplates at a confluence of ~90% for 24 h and each well of the 96-well plate was washed three times with 200 μL of PBS and test compounds were treated for 10 min. YFP fluorescence of each well was measured every 400 ms with a FLUOstar Omega microplate reader (BMG Labtech, Ortenberg, Germany). After 1 s measurement for baseline, 100 μL of 70 mM iodide solution containing 100 μM ATP was added to each well to measure ANO1-mediated iodide influx. The initial iodide influx rate determined from the initial slope of fluorescence decrease was used for measuring the inhibitory effect of test compounds on ANO1 activity.

### 4.5. Whole-Cell Patch-Clamp

HEK293T cells were transfected with pEGFP-ready tagged ANO1 24–36 h before whole-cell patch-clamp recordings of ANO1. The bath solution contained (in mM) 150 NMDG-Cl, 1 MgCl_2_, 10 glucose, and 10 HEPES (pH 7.4) and the pipette solution contained (in mM) 150 NMDG-Cl, 10 EGTA, 6.6 CaCl_2_, 1 MgCl_2_, 3 MgATP, and 5 HEPES (pH 7.2). Pipettes were pulled from borosilicate glass capillaries to have an electrical resistance of 2–3 MΩ after fire polishing. The holding potential was set as −60 mV and ramp pulses were applied from −100 mV to +100 mV in steps of 20 mV over 1 s. The pulse-to-pulse interval was 20 s. All recordings were carried out at room temperature using Axopatch 700B (Molecular Devices, Sunnyvale, CA, USA) and digitalized and analyzed using Digidata 1440A (Molecular Devices) and Clampfit 10.4 (Molecular Devices). Currents were low-pass filtered at 5 kHz and sampled at 10 kHz.

### 4.6. Intracellular Ca^2+^ Measurement

PC-3 cells were cultured in 96-well black-walled microplates and loaded with Fluo-4 NW, a fluorescent Ca^2+^ indicator, according to the manufacturer’s manual (Invitrogen, Carlsbad, CA, USA). Briefly, PC-3 cells were incubated with 100 μL of the Fluo-4 NW loading solution containing Fluo-4 in 1X Hanks’ balanced salt solution with 2.5 mM probenecid and 20 mM HEPES. After 50 min of incubation in the dark, the 96-well plates were transferred to a microplate reader for the measurement of Fluo-4 fluorescence with a Synergy Neo2 microplate reader (BioTek Instruments, Inc., Winooski, VT, USA) equipped with syringe pumps and custom Fluo-4 excitation/emission filters (485/538 nm).

### 4.7. Western Blot Analysis

Preparation of the protein sample was carried out as described previously [45]. The samples were separated by 4–12% Tris-glycine precast gel (KOMA BIOTECH, Seoul, Republic of Korea) and then transferred onto a polyvinylidene Fluoride membrane (Millipore, Billerica, MA, USA). Blocking of the membrane was performed with 5% non-fat skim milk in Tris-buffered saline containing 0.1% Tween 20 (TBST) for 1 h at room temperature. The membrane was incubated overnight at 4 °C with corresponding primary antibodies, including anti-ANO1 (ab64085; Abcam, Cambridge, UK), anti-β-actin (sc-47778; Santa Cruz Biotechnology, Dallas, TX, USA), and anti-cleaved PARP (551,025; BD Biosciences, Franklin Lakes, NJ, USA) antibodies and then was washed three times with TBST followed by 1 h of incubation with HRP-conjugated anti-secondary IgG antibodies (Enzo Life Science, Farmingdale, NY, USA) at room temperature. Visualization was performed using the SuperSignal™ Western Blot Substrate (Thermo Fisher Scientific, Waltham, MA, USA).

### 4.8. Real-Time RT-PCR Analysis

Total mRNA was isolated from PC-3 cells using TRIzol reagent (Invitrogen, Carlsbad, CA, USA). Total mRNA was reverse transcribed with random hexamer primers, an oligo (dT) primer, and SuperScript III Reverse Transcriptase (Invitrogen, Carlsbad, CA, USA). Quantitative RT PCR was carried out using Thunderbird SYBR qPCR mix (Toyobo, Osaka, Japan) and StepOnePlus Real-Time PCR System (Applied Biosystems, Waltham, MA, USA) under thermal cycling conditions of 95 °C for 5 min, 40 cycles of 95 °C for 10 s, 60 °C for 20 s, and 72 °C for 10 s. The size of the ANO1 PCR product was 82 base pairs with the ANO1 sense primer sequence as 5′-GGAGAAGCAGCATCTATTTG-3′ and the ANO1 antisense primer sequence as 5′-GATCTCATAGACAATCGTGC-3′.

### 4.9. Cell Viability Assays

PC-3 cells were incubated in 96-well microplates with growth medium supplemented with 10% FBS for 24 h before treatment with *cis*- and *trans*-resveratrol. After 72 h incubation, the medium was washed out and the MTS assay was performed according to the supplier’s protocol using Cell Titer 96^®^ AQueous One Solution Assay kit (MTS) (Promega, Madison, WI, USA). The absorbance at 490 nm was measured with an Infinite M200 microplate reader (Tecan, Grödig, Austria).

### 4.10. Wound Healing Assay

PC-3 cells were cultured in a 96-well plate to reach ~80% confluence as a monolayer. The wound was formed on a cell layer using a 96-Well WoundMaker (Essen BioScience, Ann Arbor, MI, USA) and each well was washed twice with serum-free medium. The cells were incubated with serum-free medium and IncuCyte ZOOM (Essen BioScience, Ann Arbor, MI, USA) was used to take images of the wounds. The percentage of wound closure was analyzed with IncuCyte software 2018A.

### 4.11. Caspase-3 Activity Assay

PC-3 cells were cultured in 96-well plates to reach 30% confluence before treatment of each well with *cis*-resveratrol, *trans*-resveratrol, and Ac-DEVD-CHO, a caspase-3 inhibitor. After 24 h incubation, each well was washed with PBS and incubated at room temperature with 100 μL of PBS containing 1 μM of NucView 488 caspase-3 substrate for 30 min followed by the staining of cells with 1 μM Hoechst 33,342. The fluorescence measurement of NucView 488 and Hoechst 33,342 was performed using FLUOstar Omega microplate reader (BMG Labtech) and Lionheart FX Automated Microscope (BioTek, Winooski, VT, USA) was used to take multicolor images.

### 4.12. Flow Cytometry Analysis

PC-3 cells were grown to ~50% confluence in a 6-well plate and then *cis*- and *trans*-resveratrol were treated for 48 h, then PC-3 cells were washed twice with PBS and centrifuged at 1000 RPM for 2 min. The cells were stained with propidium iodide (PI) for 15 min and then cell cycle phases were determined by using fluorescence activated cell sorting (Beckman Coulter, Fullerton, CA, USA).

### 4.13. Statistical Analysis

All experiments were performed independently a minimum of three times. The results for multiple trials were presented as the mean ± standard deviation (S.D.). GraphPad Prism 9.0 (GraphPad Software Inc., San Diego, CA, USA) was used for statistical analysis of the Student’s t-test or the one-way analysis of variance coupled with Dunnett’s T3 post-hoc test, as appropriate. *p* values less than 0.05 were regarded as statistically significant. GraphPad Prism Software was used for plotting the dose–response curve and calculating IC_50_ values.

## Figures and Tables

**Figure 1 ijms-24-01186-f001:**
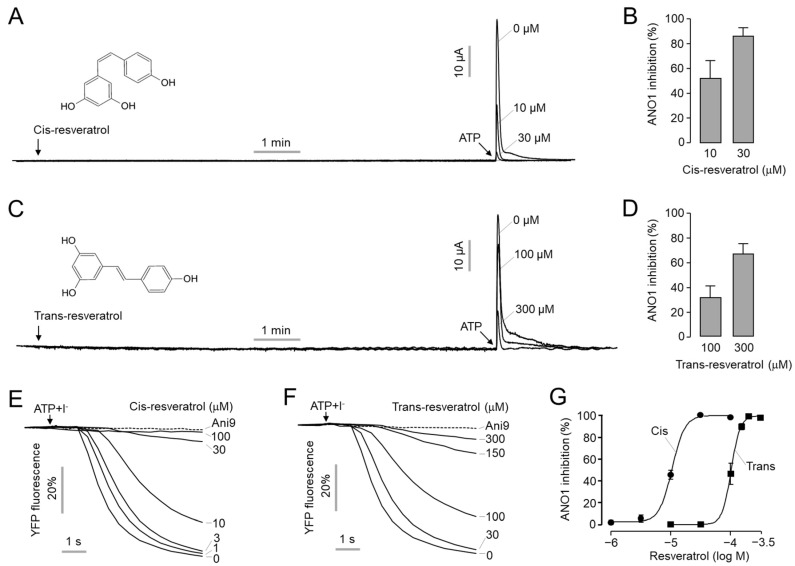
Inhibition of ANO1 activity by *cis*- and *trans*-resveratrol. (**A**–**D**) Apical membrane currents were measured in human ANO1-expressing FRT cells. Indicated concentrations of *cis*- and *trans*-resveratrol were treated for 10 min before the application of 100 µM ATP. (**B**,**D**) Summary of ANO1 current inhibition (mean ± S.D., *n* = 4–5). (**E**,**F**) The inhibitory effect of *cis*- and *trans*-resveratrol on ANO1 activity was measured with a YFP-quenching assay in FRT cells expressing ANO1 and a halide sensor YFP. Indicated concentrations of *cis*- and *trans*-resveratrol were added 10 min before the application of iodide solution containing 100 μM ATP. (**G**) Summary of dose–response (mean ± S.D., *n* = 4).

**Figure 2 ijms-24-01186-f002:**
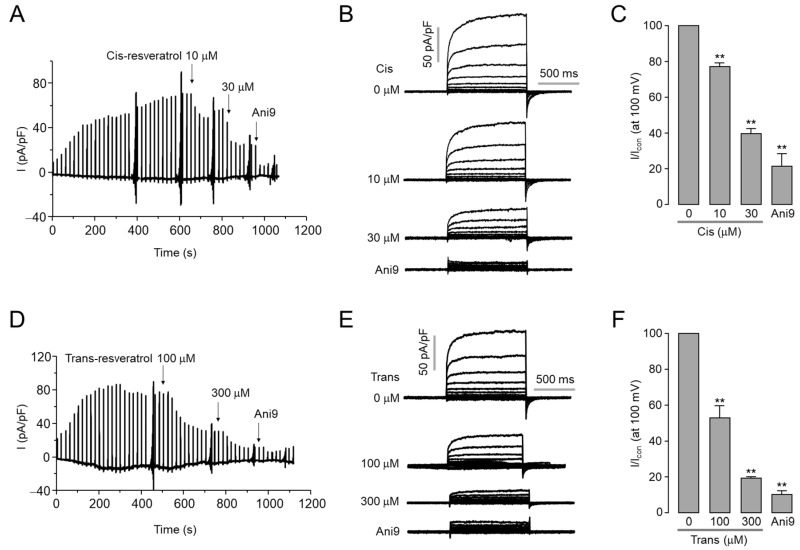
*Cis*- and *trans*-resveratrol inhibit ANO1 chloride currents. (**A**) Representative traces of whole-cell recordings of ANO1 in HEK293T cells expressing ANO1. Indicated concentrations of *cis*-resveratrol and 10 μM Ani9 were treated after the activation of ANO1. (**B**) Whole-cell currents were recorded at a holding potential of −60 mV and pulsed to voltages between ± 100 mV in steps of 20 mV. (**C**) Summary of current densities measured at +100 mV (mean ± S.D., *n* = 3–7). (**D**) Traces of whole-cell recordings of ANO1. *Trans*-resveratrol and Ani9 were treated as indicated after the activation of ANO1. (**E**) Whole-cell currents were recorded at a holding potential of -60 mV and pulsed to voltages between ± 100 mV in steps of 20 mV. (**F**) Summary of current densities measured at +100 mV (mean ± S.D., *n* = 3–7). ** *p* < 0.01 vs. control.

**Figure 3 ijms-24-01186-f003:**
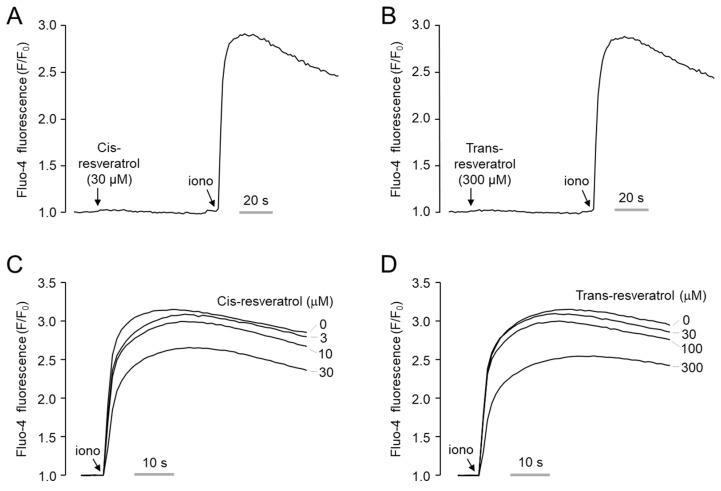
Effect of *cis*- and *trans*-resveratrol on intracellular calcium levels in PC-3 cells. (**A**,**B**) Intracellular calcium levels were measured using Fluo-4 NW. Cells were treated with the indicated concentrations of *cis*- and *trans*-resveratrol and stimulated with 10 μM ionomycin. (**C**,**D**) Cells were pretreated with the indicated concentrations of *cis*- and *trans*-resveratrol for 10 min, followed by stimulation with 10 μM ionomycin.

**Figure 4 ijms-24-01186-f004:**
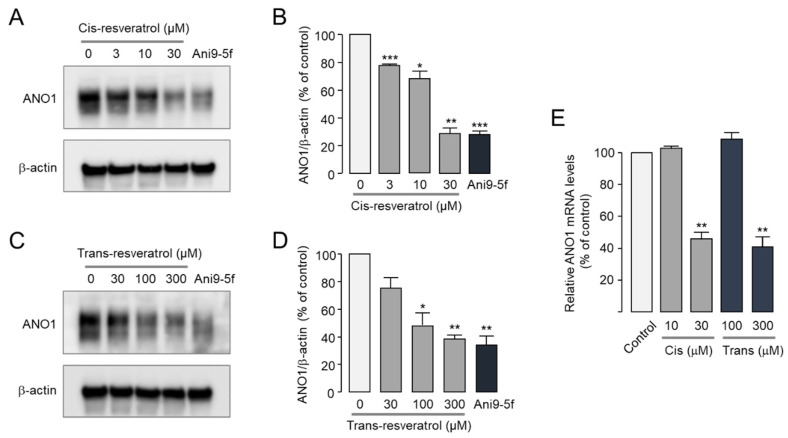
Effects of *cis*- and *trans*-resveratrol on ANO1 expression levels in PC-3 cells. (**A**–**D**) Protein expression levels of ANO1 and β-actin were measured by Western blot analysis. Cells were treated with the indicated concentrations of *cis*- and *trans*-resveratrol and 10 μM Ani9-5f for 24 h. (**B**,**D**) ANO1 band intensities were normalized to β-actin band intensities (mean ± S.D., *n* = 3). (**E**) mRNA expression levels of ANO1 were measured by real-time PCR after cells were treated with the indicated concentrations of *cis*- and *trans*-resveratrol for 24 h (mean ± S.D., *n* = 3). * *p* < 0.05, ** *p* < 0.01, *** *p* < 0.001 vs. control.

**Figure 5 ijms-24-01186-f005:**
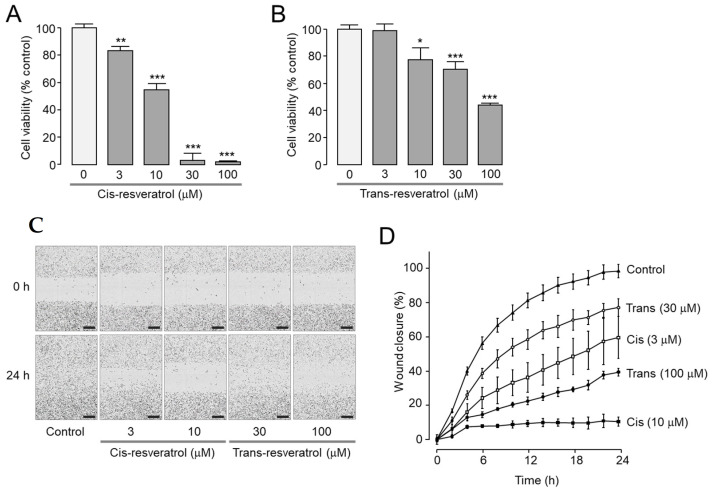
Effect of *cis*- and *trans*-resveratrol on cell viability and migration in PC-3 cells. (**A**,**B**) Effect of *cis*- and *trans*-resveratrol on cell viability. Indicated concentrations of *cis*- and *trans*-resveratrol were treated for 72 h and cell viability was estimated with MTS assay (mean ± S.D., *n* = 5). (**C**,**D**) Effect of *cis*- and *trans*-resveratrol on cell migration. Cells were treated with indicated concentrations of *cis*- and *trans*-resveratrol and the wound closure was measured every 2 h for 24 h after wound generation (mean ± S.D., *n* = 4–5). Scale bars represent 300 μm. * *p* < 0.05, ** *p* < 0.01, *** *p* < 0.001 vs. control.

**Figure 6 ijms-24-01186-f006:**
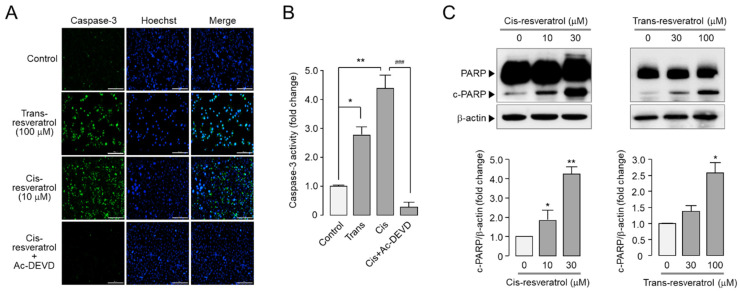
Effects of *cis*- and *trans*-resveratrol on caspase-3 activity and PARP cleavage in PC-3 cells (**A**) Images were taken after 24 h treatment with *cis*- and *trans*-resveratrol. Cells were treated with the caspase-3 substrate (green) and Hoechst 33,342 (blue) for 20 min before image acquisition. Scale bars represent 200 μm. (**B**) PC-3 cells were treated with *cis*-resveratrol at 10 μM in the presence or absence of 10 μM Ac-DEVD-CHO and *trans*-resveratrol at 100 μM for 24 h. Caspase-3 activity was measured 20 min after treatment of 2 μM caspase-3 substrate (mean ± S.D., *n* = 3). (**C**) Cells were treated with indicated concentrations of *cis*- and *trans*-resveratrol for 24 h and immunoblot analysis was used to measure the expression level of PARP, cleaved-PARP, and β-actin. Cleaved-PARP protein intensities were normalized to those of β-actin (mean ± S.D., *n* = 3). * *p* < 0.05, ** *p* < 0.01 vs. control, ### *p* < 0.001 vs. *cis*+Ac-DEVD.

**Figure 7 ijms-24-01186-f007:**
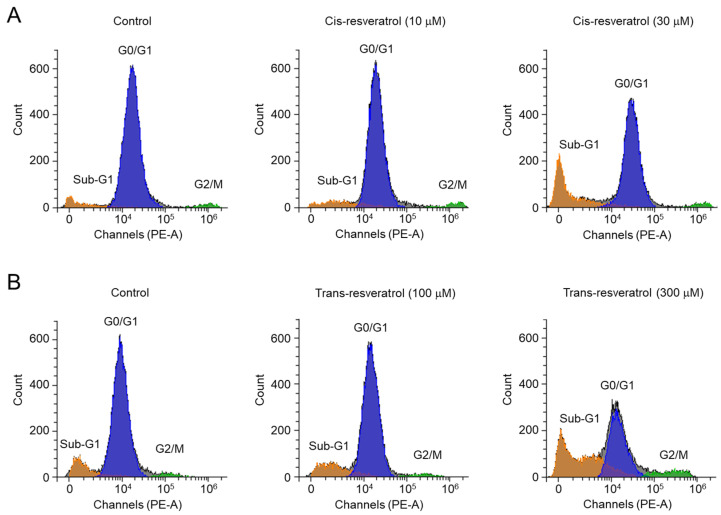
Effect of *cis*- and *trans*-resveratrol on cell cycle in PC-3 cells. (**A**,**B**) Cell cycle phases were observed by propidium iodide staining followed by flow cytometric analysis after the cells were treated with the indicated concentrations of *cis*- and *trans*-resveratrol for 48 h.

## Data Availability

The data that support the findings of this study are available from the corresponding author upon reasonable request.

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
