# Peer review of "Inhibition of ANO1 by Cis- and Trans-Resveratrol and Their Anticancer Activity in Human Prostate Cancer PC-3 Cells"

_ijms, 2023, doi:10.3390/ijms24021186_

Round 1

Reviewer 1 Report

1. line 108: why HEK293T cells rather than PC-3 cells are employed for testing?

2. Why are standard deviation and standard error used in statistical analysis? It is recommended to choose anyone, but do not mix them.

3. line 101: why is n=4-5? Did one of these values be deleted in some tests?

4. What post-hoc test was used with the ANOVA to determine differences among the treatment groups? Some experiments were more than two groups, so t-test could not be used. Using t-test instead of one-way ANOVA will increase the probability of making Type I error. Are all data normally distributed? If not, what statistical analysis method should be used? The following articles may be useful for your statistical analysis and statistical identifiers: Toxicology, 2021, 450: 152681; Chemico-Biological Interactions, 2022, 354: 109844. Thus, figure captions should be modified with the modification of statistical identifiers.

5. Fig. 5D: The horizontal coordinate lacks a title.

6. This article mainly studies prostate cancer. Why did you choose two cell lines that have nothing to do with prostate cancer (FRT cells and HEK293T cells)? And the source of the cell line should be written.

7. Suggesting to add detailed product numbers information about the antibodies in materials and methods.

8. The discussion part is not deep enough. It needs to be improved and the issues explained by different experiments need to be logically connected.

Author Response

We greatly appreciate the editor’s and reviewers’ efforts to carefully review our manuscript and the valuable comments and suggestions offered for the improvement of the manuscript (ijms-2123769). We have made each of the suggested revisions. The points of criticism raised by the reviewers were addressed by a point-by-point response. Changes in the manuscript text are highlighted in red color font.

Reviewer #1:

  1. line 108: why HEK293T cells rather than PC-3 cells are employed for testing?

Response: Thank you for the comment. HEK293T cells are widely used for the evaluation of pharmacological properties in a transient expression setting. In the patch clamp study, HEK293T cells were used because they can express ANO1 with higher efficiency than PC3 cells and can efficiently separate single cells.

  1. Why are standard deviation and standard error used in statistical analysis? It is recommended to choose anyone, but do not mix them.

Response: Thank you for the comment. As suggested, all results were expressed as mean ± standard deviation.

  1. line 101: why is n=4-5? Did one of these values be deleted in some tests?

Response: Yes, Ussing chamber assays were performed by treating each snap well with a single concentration of resveratrol.

  1. What post-hoc test was used with the ANOVA to determine differences among the treatment groups? Some experiments were more than two groups, so t-test could not be used. Using t-test instead of one-way ANOVA will increase the probability of making Type I error. Are all data normally distributed? If not, what statistical analysis method should be used? The following articles may be useful for your statistical analysis and statistical identifiers: Toxicology, 2021, 450: 152681; Chemico-Biological Interactions, 2022, 354: 109844. Thus, figure captions should be modified with the modification of statistical identifiers.

Response: Thank you for the valuable comments. Statistical analysis was performed with one-way ANOVA followed by Dunnett’s T3 post-hoc test as appropriate when t-test is not applicable.

  1. Fig. 5D: The horizontal coordinate lacks a title.

Response: Thank you for the helpful comment. The horizon coordinate is added in Fig. 5D

  1. This article mainly studies prostate cancer. Why did you choose two cell lines that have nothing to do with prostate cancer (FRT cells and HEK293T cells)? And the source of the cell line should be written.

Response: Unlike PC3 cells, FTR cells form tight junctions, so they are commonly used cells for measuring ANO1 apical membrane current in Ussing chamber studies, and HEK293T cells are commonly used for measuring ANO1 currents with high efficiency in patch clamp studies. Therefore, these cells were used for the electrophysiology experiments of ANO1. The source of the cell lines was described in the revised manuscript.

  1. Suggesting to add detailed product numbers information about the antibodies in materials and methods.

Response: Thank you for the comment. The product numbers for the antibodies were described in the Materials and methods section.

  1. The discussion part is not deep enough. It needs to be improved and the issues explained by different experiments need to be logically connected.

Response: Thank you for the helpful comment. The authors agree with the reviewer's opinion, and have modified the discussion section to make it clearer and more logical.

Reviewer 2 Report

The authors studied the effects of two stereoisomers of resveratrol on ANO1 activity and found that cis- and trans-resveratrol inhibited ANO1 activity with different potencies. Cis- and trans-resveratrol inhibited ANO1 channel activity with IC50 values of 10.6 and 102 μM, respectively, and had no significant effect on intracellular calcium signaling at 10 and 100 μM, respectively. In addition, cis-resveratrol downregulated mRNA and protein expression levels of ANO1 more potently than trans-resveratrol in PC-3 prostate cancer cells. Cis- and trans-resveratrol significantly reduced cell proliferation and cell migration in an  ANO1-dependent manner, and both resveratrol's strongly increased caspase-3 activity, PARP cleavage, and apoptotic sub-G1 phase ratio in PC-3 cells. The manuscript is well-structured and well-discussed. However, some points should be checked and corrected before it's accepted in this journal. 

Therefore, according to my comments, I recommended the publication of the paper after major revision.

[1]   ANO1 gene should be in italics.

[2]   The study's background should be clearly stated. Describe the introduction and review of the work (Please add more information).

[3]   Please speculate on the results. The discussion must improve.

[4]   Please provide the Conclusion section. The authors should add the significance of this research and its potential practical application.

[5]   The figure's quality should be improved.

[6]   The MS English needs to be improved. The article's English must be carefully checked for grammatical errors.

Author Response

We greatly appreciate the editor’s and reviewers’ efforts to carefully review our manuscript and the valuable comments and suggestions offered for the improvement of the manuscript (ijms-2123769). We have made each of the suggested revisions. The points of criticism raised by the reviewers were addressed by a point-by-point response. Changes in the manuscript text are highlighted in red color font.

Reviewer #2:

The authors studied the effects of two stereoisomers of resveratrol on ANO1 activity and found that cis- and trans-resveratrol inhibited ANO1 activity with different potencies. Cis- and trans-resveratrol inhibited ANO1 channel activity with IC50 values of 10.6 and 102 μM, respectively, and had no significant effect on intracellular calcium signaling at 10 and 100 μM, respectively. In addition, cis-resveratrol downregulated mRNA and protein expression levels of ANO1 more potently than trans-resveratrol in PC-3 prostate cancer cells. Cis- and trans-resveratrol significantly reduced cell proliferation and cell migration in an ANO1-dependent manner, and both resveratrol's strongly increased caspase-3 activity, PARP cleavage, and apoptotic sub-G1 phase ratio in PC-3 cells. The manuscript is well-structured and well-discussed. However, some points should be checked and corrected before it's accepted in this journal.

Major Concerns

  1. ANO1 gene should be in italics.

Response: Thank you for the comment. The ANO1 gene was changed to italics.

  1. The study's background should be clearly stated. Describe the introduction and review of the work (Please add more information).

Response: Thank you for the comment. The introduction part has been updated to better explain the study’s background.

  1. Please speculate on the results. The discussion must improve.

Response: Thank you for the helpful comment. The authors agree with the reviewer's opinion, and have modified the discussion section to make it clearer and more logical.

  1. Please provide the Conclusion section. The authors should add the significance of this research and its potential practical application.

Response: Thank you for the comment. The conclusion part has been added in the discussion section.

  1. The figure's quality should be improved.

Response: Thank you for the comment. The quality of figures has been improved by adding further clarification in the figures.

  1. The MS English needs to be improved. The article's English must be carefully checked for grammatical errors.

Response: Thank you for the comment. The English in the manuscript has been carefully checked and corrected.

Round 2

Reviewer 1 Report

Accept in present form

Reviewer 2 Report

Requested corrections were completed.